# Fisibilillah: Labor as Learning on the Sufi Path

**Youssef Carter**

Department of Religious Studies, University of North Carolina at Chapel Hill, Chapel Hill, NC 27599, USA; youssefcarter@unc.edu

**Abstract:** At the core of this study of spiritual empowerment and Black Atlantic Sufism lies the pre-occupation of understanding precisely the manner by which particular Muslim subjectivities are fashioned within the bounds of the Mustafawi Sufi tradition of religious cultivation through charitable giving and community service in Moncks Corner, South Carolina. This article examines Black Atlantic Muslim religiosities and argues that West African Sufism in diasporic context—which draws upon nonwestern theories of the body and theories of the soul—can be theorized as a philosophy of freedom and decoloniality. In the American South, spiritual empowerment becomes possible through varying forms of care and bodily practice that take place in a mosque that is situated on a former slave plantation. Meanwhile, that empowerment takes place through discourses on Islamic piety and heightened religiosity in a postcolonial Senegal. Spiritual empowerment occurs, as I show, through attending to the body and spirit as students connect themselves, via West African Sufism, to a tradition of inward mastery and bodily discipline through philanthropic efforts.

**Keywords:** labor; Sufism; African diaspora; knowledge transmission; black Atlantic; Senegal

## 1. Introduction

*"So her Lord (Allah) accepted her with goodly acceptance. He made her grow in a good manner and put her under the care of Zakariya (Zachariya). Every time he entered Al-Mihrab [the prayer chamber] to (visit) her, he found her supplied with sustenance. He said: "O Maryam (Mary)! From where have you got this?" She said, "This is from Allah." Verily, Allah provides sustenance to whom He wills, without limit."*

(Qur'an 3:37; translation via Sahih International)

Shaykh Arona Rashid Faye Al-Faqir always cautions the students under his care to realize that ones' sustenance (*rizq*) comes from Allah only. The believer, therefore, should not fear impoverishment or any depletion of wealth due to giving charity. His manner of teaching is not only predicated upon, explicating how Maryam, mother of Sayyidina Isa (Jesus the Christ), was protected and provided for directly from God's endless bounty, but also the Shaykh's pedagogical approach involves modeling trust in God through embodying an ethos of giving and service. Therefore, as his students sat encircled around him listening attentively to his teachings in the Moncks Corner mosque—many of them beneficiaries of Shaykh Arona's commitment to charitable service at one time or another—his points were made that much clearer as they watched him embody service through organizing and facilitating the dispensation of food. In between unfolding accounts of Maryam's pious character, Shaykh carefully and assertively directed some of his students (*taalibes*) in ensuring that the preparations for the community dinner were being carried

out smoothly. Just as was the case for more than two decades in Moncks Corner, the *fuqara*[1] gathered at the mosque on Sunday afternoons to participate in its community potluck. Although individual dishes are mostly cooked by the women, all members who are old enough take it upon themselves to offer their assistance in cleaning and perfuming the mosque, moving chairs, arranging food and drink items on the folding tables, and clearing space for guests. Every Sunday morning, such attentiveness to giving and service is but an example of how Islamic piety is embodied in the *zawiyah* of Moncks Corner. Even still, Shaykh Arona Faye makes sure that his students recognize where true success and power are to be found.

This article examines the significance of a group of predominantly African and African-American Sufi Muslims who have sought inward refinement and retreat from within the Mustafawi Sufi tradition that originates in Senegal. Over the course of my ethnographic research at the *zawiyah*-mosque[2] in Moncks Corner, South Carolina, and a number of research visits to Senegal, I have had an opportunity to think more deeply about spiritual empowerment, and further, how the people that I have learned from, and with, imagine and articulate empowerment for themselves. I ask these questions in the spirit of trying to think through how knowledge is articulated and transmitted in the Mustafawi tradition that has communities of students in both West Africa and in the American South. In addition, I am concerned with how discourses of labor and giving are enacted amongst African and African-American Muslims who engage themselves in the Mustafawi tradition of spiritual expansion.

The Mustafawi Tariqa is a transatlantic Sufi Order that originated in Senegal by way of the late Shaykh Muhammad Mustafa Gueye Haydara (d. 1989) and established a presence in Moncks Corner, South Carolina, in 1994, due to the efforts of Shaykh Arona Faye, leader of the *fuqara* in the USA, and Umm Aisha Faye, an African-American Muslimah and local matriarch of the Moncks Corner *zawiyah*. It is through the Mustafawi Tariqa that transatlantic solidarities are configured in the small, blue-collar town and beyond. In order to discuss the cultivation of religious selves among Muslims of African descent in that space, I analyze how discourses of piety and liberation are framed as African-descended Muslims use their bodies to perform and internalize Islamic knowledge through varying kinds of pious labor that contribute to the materializing of diasporic identities. I also explore how those bodily performances play a part in the forging and maintenance of transatlantic religious relationships (Lovejoy 1997; Holsey 2004; Clarke 2004; Matory 2005; Griffith and Savage 2006; Cohen and Hear 2008; Garbin 2013). In order to interrogate how pious formation is activated through bodily dispositions and performances, this analysis highlights how and by what means a postcolonial West African Sufic tradition of moral–ethical training is deployed to address the needs of African-American Muslims in the United States. Labor, in this case—cooking, praying, completing domestic tasks, fundraising, and religious performance—acts as a vessel for religious instruction amongst and between two distinct groups of diasporic Africans that possess variant relationships to the American South and Senegal. I offer, therefore, that these performances amongst African and African-American Muslims are informed by spiritual and historical reconnections that are grounded in affective diasporic relationships between the American South and Senegal—a discursively imagined homeland as it emerges in the Moncks Corner Mosque (Yamba 1995; Alpers 2000; Clarke 2004; Ho 2006; Kane 2011).

---

[1] As in other Sufi traditions, Shaykh Faye has named his community the "*fuqara*", which not only connotes dependency on God, it also functions to mark those who similarly identity as spiritual companions regardless of their location. Conceptually, the name "*fuqara*" is sourced from the Qur'an and is used by Shaykh Arona Faye to express an utter and complete dependence on God. For example, the Qur'an states " . . . *If they are poor, God will provide for them from His bounty: God's bounty is infinite and He is all knowing*" (Qur'an 24:32). Shaykh Faye has taken its singular form as a name ("*Al-Faqir*") to recognize his own dependency in relation to God and has named his students/followers in a similar fashion to signify their respective dependence.

[2] *Zawiyah* is an Arabic word that can be translated as "corner" or "nook"; its connotation is that of a place of refuge—a location where Sufi disciples assemble and focus on spiritual growth and religious education in community with other Sufis. In other Islamic contexts, they can be referred to as *khanqahs* or *dahiras*. In South Carolina, the *zawiyah* of Moncks Corner is the mosque, and in Dakar, it was a large house occupied by local family members, students and guests.

Currently, the Mustafawi Tariqa maintains small and moderately sized collections of students (ranging from 20 to 150) in a number of spaces that include major American cities such as Philadelphia, Washington, D.C, and Atlanta, as well as members in Senegal, Gambia, Mauritania, Indonesia and in Spain. In the United States, the majority of students who have entered the Tariqa and have declared allegiance (*bayah*) to Shaykh Arona Faye are African-Americans—Shaykh refers to himself, them, and all members of the Mustafawiyya as "*fuqara*"—who have inserted themselves religiously, culturally, and pedagogically into a West African Sufi tradition which emphasizes religious study and the practice of *dhikr* (remembrance of God). Shaykh Arona Faye, whose name has been alternatively spelled as "Sheikh Harun" in other writings, is a Senegalese scholar who relocated to the southeastern region of the United States from West Africa to spread the religion of Islam and expose American Muslims to the rich tradition of spiritual purification and righteousness. At the same time, many of those who are African-American members of this tradition have made it a point to either travel to Senegal themselves to strengthen transatlantic ties with West African compatriots and visit sacred burial sites in the small city of Thiès, or have sent their children to study Qur'an for a number of months or years at the now-defunct Fuqara International Islamic Academy in Dakar, Senegal.

In spite of the many spaces where Mustafawiyya inhabit, two primary locations—Thiès, Senegal and Moncks Corner, South Carolina—act as centers of pilgrimage for Mustafawiyya members. The North American center of the Tariqa is located in Moncks Corner, South Carolina, which is a small southern town that has a population of less than 9000 residents and is about 7.5 square miles in size. All religious holidays, celebrations, and significant events amongst the Mustafawiyya in America occur in Moncks Corner and most American members travel there from throughout the eastern US at least once a year. Meanwhile, Thiès is the third largest city in Senegal, lies about 70 km east of Dakar, and has a population of less than 400,000 residents. This city, considered a gateway and transport hub to Dakar and other neighboring cities, is the point of origin and West African center of the Mustafawiyya network. This is also where Shaykh Arona Faye and other prominent leaders in the Tariqa were born. Additionally, the founding leader (Shaykh Mustafa), Faye's mother, and beloved grandfather are all buried there. While Moncks Corner acts as the primary location in which this research was situated, Thiès was the secondary location where research took place.

## 2. An Economy of Giving in the Black Muslim Atlantic

The African-descendant Muslim subjectivities that I describe here are cultivated within the bounds of a specific practice of spiritual transformation that is mediated by inclusion in the Mustafawi Sufi tradition. This analytical framing of West African-derived Sufi logics, therefore, extends beyond studies of Muslim ethical cultivation via embodiment in Muslim majority contexts (Mahmood 2011) and in diaspora (Jouili 2015; Eisenlohr 2006) by examining how ethical cultivation operates as a work of self-healing in the context of racial violence and oppression. In the American South, that cultivation centers around the mosque that is situated on a former slave plantation and is informed in this case through *tarbiyah* (moral training/alchemy of the inward self)—of which the religious poetry of Shaykh Mustafa Gueye is a part—and the call to actualize a return to the *fitra* (original state of humanity inclined toward God-consciousness and cleansed of negative or harmful experiences) via *tazkiyya* (eradication of the lower, race-inflected Self/disfigured human). Remembrance of Allah involves, and is contingent upon, an intimate awareness or discovery of one's higher Self that is achieved through attending to the body and spirit. It is formed from a theory of the body that argues that corporeal discipline can positively impact the soul. Students connect themselves to these West African Sufi practices because the processes of *tarbiyah* and *tazkiyya* provide a foundation for addressing traumas produced through enduring bodily and historical violence. In Dakar and Thiés, however, the performance of spiritual cultivation takes place in the homes of the Mustafawi adherents. Through an emphasis on inward mastery and bodily discipline through charity and service

(*khidma*), they also practice an approach to psychospiritual care that is derived from an Islamic epistemology. Healing and spiritual expansion are achieved as well through reading the spiritual poetry of Shaykh Mustafa, hosting guests and committing themselves to embodying piety through serving their teacher. At the same time, however, students witness the direct forms of care dispensed and practiced by their teacher as he attends to the material and practical needs of community members and visitors. I also examine the multiple kinds of shared labor in which both men and women personally attend to the Shaykh's needs, attend to guests, clean the mosque, and prepare elaborate dishes for visitors and other mosque members. These kinds of labor are interpreted as motivated by the gratitude/indebtedness that are generally felt throughout the Tariqa for the quite visible labor of Shaykh Faye. Moreover, this analysis extends beyond a focus on material exchanges by studying the manner in which charitable giving is mobilized and articulated as a bodily devotion. Modeling care in a communal capacity is thus a vital and vigorous component of moral training.

In examining the various kinds of labor as a bodily practice through which Islamic piety is displayed in and throughout the Mustafawi spiritual network, it is necessary to illustrate how that knowing is embodied. Therefore, I use the term "alchemy"—a popular term among Sufi groups—as a manner of describing the programmatic, scientific transformation of objects, which carries with it connotations of spiritual growth and expansion, as well as the transformational relationship between African-descended students and a West African Sufi tradition of spiritual care.[3] I deploy this term to suggest the transformative relationship that emerges as African-descendant Muslims, in particular, learn to dissolve their egos (*tazkiyyat-ul-nafs*) and attain a higher sense of Islamic piety. Moreover, this process of transformation includes, according to the ethnographic data that I have collected, the cultivation of African diasporic religious identities via the application of secretive prayers, devotional performances in concert and in solitude, and journeys taken to pay homage to their shuyukh in an imagined spiritual homeland. If alchemy is the transformation of matter, the goal of the transformation I describe here is a matter of transforming hearts. This transformation is achieved, however, through disciplining the bodies of Muslims of African descent in the American South and beyond.

Benjamin Soares shows gift-giving among Sufi adherents and the prestige garnered from lavish incentives for various kinds of spiritual care by understanding prayer as a medium of exchange in West African Muslim contexts, and I draw upon his insights by including how various forms of labor and service make spiritual intimacy between teacher and students possible (Soares 2005). I, therefore, complement an emphasis on the way in which power and authority configure the interpersonal relationships amongst the *fuqara* and, meanwhile, endeavor to make sense of how labor is perceived and theorized amongst members of *Masjidul Muhajjirun wal Ansar* (the mosque in Moncks Corner). For example, the proximity that students not originally from the Moncks Corner region is, in many cases, directly made possible by participation in itinerant, contractual labor. Moncks Corner is primarily a blue-collar town where regular and gainful employment can be difficult and, consequently, many of the male members of the *zawiyah* choose to engage in contractual labor as electricians for which they must work in other locations around the country in order to funnel monies back to their respective households. For many of the households who are part of the Moncks Corner *zawiyah*, there is at least one member (generally the male head of household) who engages in the kind of electrical work that requires travel to locations, ironically, often far away from the *zawiyah*. These relocations are temporary, with longer contracts sometimes cut short by the at-will resignations of the Muslim men who have left town in order to make money for their families and or provide financial support for the Mosque, visit wives in Senegal, or to participate in the annual Shaykh Mustafa Day

---

3 Literature on Sufism and works by notable Sufi masters are replete with references to alchemy. For example, see "*The Sufi Message of Hazrat Inayat Khan: The Alchemy of Happiness*" or Waley's "*Sufism: Alchemy of the Heart*". In fact, Shaykha Maryam Kabeer Faye, devoted student of Shaykh Arona Faye, has penned a memoir of her own spiritual journey that is entitled "*Journey Through Ten Thousand Veils: The Alchemy of Transformation on the Sufi Path*".

in Thiès (Senegal). I understand these choices as informed by a logic not merely configured by local economic conditions. The decision by the *fuqara* men to work as electricians and preferring contractual labor that requires, in many cases, much time spent in different parts of the country affords their families the ability to remain in Moncks Corner in close proximity to the religious community that sustains them. More importantly, it seems, the *fuqara* more easily remain physically connected to their teacher and it is through that labor that the transmission of Islamic knowledge through a specific West African Sufi pedagogy is dispensed.

At the core of the relationship that allows for the patronage of the teacher via the disciple is the import of religious authority. This notion has been studied by scholars of Islam in Africa by framing the allure of religious authority as "charisma" or "charismatic authority"—a notion taken from Max Weber's insights on religious authority (Weber 1968). Those who have studied this phenomenon agree that what facilitates the exchanges (seen and unseen) between teacher and disciple is the power imbued from within the "miracle-working founder" (Cruise O'Brien and Coulon 1988) who is often posthumously recognized as a saint. This sainthood, especially when connected to a series of successive teachers and shuyukh, provides the spiritual legitimacy upon which charisma is built. Around this, the institution of the spiritual network/Sufi Order (*tariqa*) is constructed and expands from the emergence of a "routinized charisma". As Donal Cruise O'Brien reviews in his introductory chapter in the edited volume, "*Charisma and Brotherhood in African Islam*", he questions the universal applicability of Weber's position that an environment of crisis is a prerequisite for the development of the social organization around which charisma is legitimated.[4] For example, if this were the case, then the jihads of Shaykh Usman dan Fodio (alt: Uthman b. Fudi) in response to the emerging slave trade and religious syncretization that threatened the integrity of an orthodox Islam in West Africa provide a backdrop to the emergence of commercial agricultural pursuits that accompanied the rise of charismatic authority figures in the region (e.g., the groundnut enterprise among the Muridiyya of Shaykh Ahmadu Bamba (d. 1927). On the other hand, it is perhaps more likely that the success of any significant religious enterprise necessitates an economic venture which would consider the practicalities of sustaining any sizable social organization or religious movement. Just as yearning hearts require knowledge, hungry bellies require food. While the notion of "crisis" as requisite to the rise of charismatic leadership is perhaps debatable, the argument is compelling in considering the example of the Mustafawi in South Carolina. Yet, again, it is more likely that their training and participation in the electrical trade (via contractual labor) among the Mustafawi is an economic strategy that is the result of residence in a region with little to offer in the way of lucrative employment. Of course, the internal spiritual discourse produced from within the *zawiyah* characterizes discipleship as the recognition of some spiritual deficiency in the devotee (i.e., some personal crisis) that has guided them to their teacher, but this, in my reading, is not necessarily how Max Weber articulated the allure of charismatic authority. On the other hand, the development of a proper "prayer economy" requires participants who can be of benefit to each other. In order for this symbiosis to be possible, those living in Moncks Corner must be able to take advantage of what Shaykh Faye has to offer. Moreover, in order for the aspirant to be able to participate in the industry of spiritual power—to purchase prayers—to assist in the purchase and upkeep of the mosque, s/he must be able to afford it via some means of regular or sustained employment.

---

4　Donal Cruise O'Brien also notes "racial confrontation" to be a considerable factor in the allure of charismatic leadership and the emergence of the tariqa as a local configuration of popular Islam in Africa, particularly among those who were made vulnerable within racial hierarchies created by colonial intrusions (whether European or Arab). While the Muridiyya of Senegal and the Hamallists of Sudan emerged from within a French colonial context, the Qadiriyya of Tanganyika distinguished itself from the Swahili and the coastal Arabs. See "*Charisma and Brotherhood in African Islam*" (Cruise O'Brien and Coulon 1988, p. 21). This notion provides an interesting point in considering the discourses of self-determination and cooperative economics among African-American Muslims in a West African Tariqa (*Mustafawi*), all housed in a mosque situated in an environment profoundly impacted by histories of racial confrontation and disinheritance. At the same time, the process of financing the mosque without seeking to rely on South Asian or Middle-Eastern financiers also provides interesting commentary on contentions between Muslim communities in the United States.

In the same volume, Murray Last (1988) discusses the historical emergence of charismatic force in northern Nigeria (Kano) as an alternative strategy of power-broking that compelled ordinary actors and leaders alike to seek success through patronizing the medicinal and spiritual efficacies of religious masters who had the ability to assist in preventing military defeat. In nineteenth-century Kano and prior, medicines manufactured by Muslim scholars were utilized by kings to ensure their ability to retain physical and mystical power over their enemies. Thus, "*charismatic power*" is understood in this context as a "force of personality in contrast to the force of arms" or, in other words, mystical power was dispensed through various forms of patronage and was utilized as a different form of coercion—another form of weaponry that was used in conjunction with violence (Last 1988, p. 186). "Prayer economy" in Kano, as it was introduced through Last, developed alongside the Nigerian oil boom of the 1970s as African political leaders sought to have Muslim scholars pray for their continued success on their behalf. This direct funding of Qur'anic expertise led to the proliferation of traditional Qur'anic students who also sought their own monetary compensation in exchange for the labor of surrogate supplication financed by affluent patrons. While Last does not appear to explicitly define "prayer economy" in such a way that includes the purchase of medicines and healing from Sufi experts, it is implied throughout the broader discussion of services rendered in-kind, either for the dispensation of blessings (*barakat*), miracles (*karamat*), or for access to the mystical power, that the Shaykh possesses and utilizes on behalf of the patron.

Benjamin Soares (2005) discusses how histories of colonization and spiritual authority intersect in Nioro (Mali) in order to shed light on the manner in which "Islamic esoteric sciences" are employed to navigate Nioro social landscape. In Soares' account, these sciences include "petitionary prayers and blessings, instruction or guidance in alms-giving, geomancy, mystical retreat, decision-making via divine inspiration, the confection of written texts . . . , astrology and medicine" (Soares 2005, p. 127). Ultimately, Soares' project is to outline how power and authority shift throughout Nioro's colonial and postcolonial periods as charisma tended to centralize upon particular figures in the religious landscape. This centralization of power gave rise and led to the emergence of saintly lineages. Benjamin Soares defines the term "prayer economy" by placing emphasis on processes of commodification and frames it as a phenomenon that vastly shifts relations between saints and followers:

> "The prayer economy is, in effect, an economy of religious practice in which people give gifts to certain religious leaders on a large scale in exchange for prayers and blessings. I argue that certain processes of commodification—the exchange of blessings and prayers for commodities, the proliferation of personal and impersonal Islamic religious commodities—have proliferated and intensified around such religious leaders in the postcolonial period. Such processes of commodification have helped to transform the relations between religious leaders and followers. In fact, they have facilitated the personalization of religious authority in certain Muslim religious leaders with reputations as saints, to whom many ordinary and elite persons have turned for [succor]. That is, religious authority has come to be centered on a few individuals rather than institutions like the Sufi orders with which they have historically been associated". (Soares 2005, p. 153)

Inside of this framing of the power relationships between spiritual leaders and their followers, Soares draws upon Weber's notion of the "routinisation" of charismatic authority to examine how saints in the modern period command prestige (Weber 1968). It is lineage that provides this command to saintly power. As I witnessed during my time spent around Shaykh Faye in South Carolina and in Senegal, his own prestige is intimately tied to his lineage—being the premier student of Shaykh Mustafa Gueye (1926–1989) and the first grandson of Shaykh Mustafa's father, Shaykh Samba Gueye (1879–1966), as well as a descendant of Prophet Muhammad through his mother. However, his prestige is also

made firm and communicated to potential followers and new students through a reputation of generous giving and spiritual care.

### 3. Fisibilillah: Giving for the Sake of God

Much has been written regarding gift-exchange and the manner in which certain forms of gift-giving operate as routes for the strengthening of social relationships (Mauss 1954; Appadurai 1988; Carrier 1991; Humphrey and Jones 1992). Time spent within the Moncks Corner *zawiyah* has urged me to consider how these exchanges resemble gift-giving practices that serve to fortify community bonds and the spiritual kinship relations between teacher and student. Ousmane Kane (2011) discusses the manner in which a transnational spiritual network provides the infrastructure for which charismatic West African shuyukh travel to Western states from Senegal and back again. There are differences, according to Kane, between tangible benefits and intangible rewards for both patrons and the shuyukh that they patronize for their services. I offer that it is the intangible rewards that are perceived as more valuable—or rather, beyond the level of value insofar as the perceived benefit is immeasurable—by participants in a prayer economy. Thus, a student who is a dutiful follower of Shaykh Faye that pays for some secret prayer (*siru*) or offers to pay for an item that Shaykh Faye owned (a red wallet, small bottle of oil, his car, etc.) should be understood as a transaction that operates beyond the level of mere purchase, or as more than some mutual exchange between equivalent parties. It is important to note that it is also necessary to investigate and offer up for analysis the multiple meanings that such transactions carry in hopes to more fully understand how meaning is contrived from the possession and exchange of materials among initiates and between Muslims on the path of spiritual development.

While sitting in the dining hall of Masjidul Muhajjirun wal Ansars in Moncks Corner, Shaykh Arona Faye explained to me that a red notebook that he had in his possession was going to be a part of a compilation of secretive knowledge pertaining to numbers and astrological signs that he intended to bestow to one of his most trusted students.[5] According to Shaykh Faye, a student purchased this knowledge from him and it was also explained to me that there would be only three students to whom the knowledge would be sold. The book, which was still in the process of being copied by hand on behalf of his student, was a simple red journal that would eventually be one of three books which would discuss and explain numerology and its inner workings. The knowledge would be used for determining the inherent personality traits of people according to their date of birth and other factors, from which the knowledge-bearer could assist others who might consult him for advice on healing maladies, determine compatibility with a potential spouse, and prescribe prayers for specific needs. Once the knowledge was bestowed, or successfully transferred, to the student, Shaykh Faye acknowledged that his student would be able to use it to benefit himself financially while also providing benefit to his patrons. Such dispensation of secretive knowledge is, of course, only sold to trusted students who have displayed a high and constant appreciation for Islamic etiquette (adab). Once they have gained the confidence of the Shaykh, only then do they become eligible to possess invaluable knowledge and secretive prayers.

Those within listening distance of a bard who devotes his time to singing the praises of Allah and His Prophet offer to reward him with small generosities do so because they recognize that the bard provides benefit to his audience. Renumeration is, in most cases, voluntary and given at the discretion of the giver. Likewise, a student who pays her or his Shaykh for specific formulaic prayers that are used for one's increase in health, income, or knowledge, in most cases, does so because the student understands that her or his income that touches the hand of the shaykh is blessed. According to the logic of exchange in the Mustafawi tradition (and Sufi ways of knowing more generally), there is no properly mutual exchange in currency for service between the shaykh and his student. Shaykh Faye

---

[5]   Interview, Shaykh Arona Faye, 28 November 2016.

carries out this labor, and asks for renumeration from his students, not so much for income it seems, but rather, in order to provide a sustainable means of redistributing monies toward less-fortunate people within his care or, in some cases, to pay for much-needed repairs to the mosque. Monies garnered always seemed to be dispensed and redistributed on behalf of those in need within the Moncks Corner community first and then outward. At the same time, though, such visible gestures surely add to his prestige, especially when in a location where economic conditions are difficult, e.g., in Senegal and in Moncks Corner (USA). Shaykh Faye's reputation for giving is known throughout Moncks Corner, and even more so, in Senegal. Upon our visit to Dakar and Thiès for the Shaykh Mustafa Day Celebration in January 2015, Rasheed P., Luqman D., and I were given a small apartment in HLM Grand Yoff to share. The three-bedroom condo was a five-minute walk from Ndey Faye's small house nearby where Shaykh Faye lodged. It was not long until the neighboring community learned that he was in town. As a result, many visitors began to patiently sit in the front room while waiting to meet with Shaykh Faye about a week into our visit. During our drives between Dakar and Thiès, Shaykh Faye revealed to the small group of men that most of the local guests who have visited the house in order to see him were there to ask for money. Shaykh Faye has explained that he loves to give, and that such adoration for generosity is couched firmly in the prophetic tradition.

Of course, from the outside looking in, such requests for money from working-class disciples on the part a highly revered spiritual leader can too easily present as a kind of exploitation. It must be explained, however, that my discussions with members of the Tariqa have revealed a logic in which there is a trust that any monies devoted to charity will be dispersed more effectively by the Shaykh to those in need. In practical terms, offering charity through the Shaykh allows those in need to maintain their anonymity and, especially in Senegal, the charitable can avoid being constantly solicited by those in need. During a candid conversation with Hassan Faye, a son of Shaykh Arona Faye, in his Dakar condominium, it was impressed upon me that his father differs from most other shuyukh because while most could be described as generous, it is well known that the Shaykh regularly gives away his own ornate West African-style grand boubous to students and friends.[6] In this way, his generosity exceeds the general custom in which one would, according to Hassan, normally refrain from giving one's clothes away for fear that it might be used to spiritually harm the owner. This manner of betrayal of the Senegalese cultural norm I understand as a pedagogical device for stressing to his students the importance of giving as well as a strategy for distributing wealth to community members with less means. Among the myriad descriptions that Shaykh Arona shares with his students about his own teacher, Shaykh Mustafa Gueye (d. 1989), is one in which he describes an instance in which he was traveling some short distance in the city of Thiès with Shaykh Mustafa. Confused as to why Shaykh Mustafa elected to take a taxi when simply walking would be just as convenient, Shaykh Mustafa instructed Shaykh Arona to consider that the money given to the taxi driver for his services rendered would then be used to feed the driver's family. Similarly, the practice of giving away his tailored clothes to friends and students creates a consistent means for Shaykh Arona Faye to solicit the expertise of his trusted tailor in Senegal, Serigne Mbaye, who uses those funds to feed his own family. At the same time, giving away beautiful clothing provides others with an opportunity to wear, or embody, the pious prestige and confidence that beautiful clothing brings. Not only does giving away money and valuable items encourage the redistribution of wealth; the ability to witness the careful and nonchalant dispensation of money and valuable items to the less fortunate models pious action for the students of Shaykh Faye.

Halfway into our stay in Senegal, Shaykh Faye revealed to some of us that he'd given away all the money that he'd brought. He subsequently requested that we each visit an ATM and do our best to give him what we could so that he would be able to continue

---

6　During my most recent trip to Senegal, I learned that the brown and blue grand boubou that Shaykh Arona Faye gifted to me was quite similar to the one he wore during his son's, Hassan, wedding to Rukayah Nurridin the previous year.

assisting those who visited him. The logic engrained in such a request was that since we were more privileged than his guests, we had a duty to distribute as much alms as we could to those who were in need. While such a transaction involved providing money to the Shaykh to support his dispensation of alms to his visitors (thus adding to his already heightened prestige), it should be recognized that those *fuqara* who willingly give their money and time to serve the Shaykh do so with the understanding that such service will result in spiritual benefit. Such provision is part and parcel of the training of the *faqir* on the path of inward refinement. It is precisely through monetary sacrifice that prayers are rendered more efficacious. Furthermore, as the *faqir* provides material support for the Shaykh, the Shaykh will also count that *faqir* as among the foremost and convicted of his students. At the same time, the *faqir* desires to be viewed as an aspirant by his/ her teacher. The one who reaches into his/her own pocket for the sake of others knows that such an activity provides its own reward because charity for the Muslim is a source of benefit unseen. Not only is the *faqir* being provided for, when one views this exchange as spiritually beneficial for the dispenser of alms, it is also a means of pedagogy whereby the *faqir* witnesses the care with which Shaykh dispenses alms to those in need. In this way, he models a reliance on the otherworldly and, as well, a de-emphasis of the import of material wealth. Secondly, the Shaykh continues to instruct his students with regard to the better etiquettes of a true believer. Thus, we must understand and analyze such exchanges of funds and service as willing modes of learning in addition to the benefits provided on behalf of religious authority.

The conclusion could also be made that the prestige gained by being a saint (*wali*) or a charismatic shaykh leads to the tangibility of rewards enjoyed by such actors in the way of material possessions. Yet, what I have witnessed is that such prestige, especially when placed within challenging economic conditions, seems to be a self-reinforcing problem. The more prestige a shaykh has, the more frequently he will be visited by those who are in need of material assistance. Further, the more material assistance one provides, the greater his prestige will be. To be fair though, unlike most other shuyukh, Shaykh Faye freely dispenses money and clothes to students and admirers throughout his network. During the group's stay in Senegal for the 2015 Shaykh Mustafa Day celebration, Shaykh Faye made it a point to explain to our group as we were witnessing the difficulty he faced with keeping up with such requests for assistance, that too often people cannot distinguish between generosity and wealth. He quietly explained that many of those that he helps assume that he is wealthy because he gives so much.

At the same time, this does not mean that Shaykh Faye does not engage in strictly transactional exchanges when clients come to him for special requests to treat ailments that Western medicine cannot rectify. On one occasion as a relatively small group of his students were gathered at his house for dinner, Shaykh Faye recounted an instance in which he was hired by a South-Asian (or Arab) Muslim family to rid their house of a *jinn* (spirit) that had taken up residence in their house and possessed one of the women in the household. He mentioned that the family had solicited the services of another shaykh who could not manage to fix their problem. That shaykh then, knowing about Shaykh Faye's spiritual prowess, contacted him to step in and help the family. These external transactions that occur do not seem to take place in order to strengthen spiritual bonds between Muslim actors. They more so take place based on the reputation and prestige of the Shaykh, and less so on an intent to maintain some sustained spiritual cultivation of the client. Through time spent within the *zawiyah* of Moncks Corner and witnessing the multiple exchanges that took place, I have concluded that transactions that did not place an emphasis on long-term care and spiritual pedagogy mostly occurred with clients who were not among the *fuqara* of Moncks Corner.

The more devoted students of Shaykh Faye—those who could be properly referred to as "students" or *taalibes* (as opposed to followers, admirers, or mere community members)—tended to purchase the more expensive items that involve more labor from their teacher. These items are also understood as investments, particularly by those students who have

been awarded a "mantle" or cloak that marks them inside the community as having met certain criteria that includes memorizing at least the last thirtieth portion of the Qur'an (*juz amma*), forty or more hadiths in Arabic and English, and the ninety-nine names of God ('*asma al husna*) and their meanings, in addition to a continued path of growth. Additionally, being given a mantle is usually a process that is done in front of the community during some gathering, and it is witnessed that this recipient both has the confidence of carrying on the aims of the Tariqa while also marking them as a figure deserving of heightened respect by the Moncks Corner community. Abdur-Rasheed W. is one of these figures who has met the Shaykh's criteria for teaching others and has been awarded with a mantle. Devoted students such as Abdur-Rasheed are driven to engage in the attainment of more advanced and intricate knowledge from Shaykh Arona Faye due to their desire to carry on his legacy in more formalized manner. "Mystical apprenticeship", a term I borrow from Joseph Hill's study on female spiritual authority in Senegal (Hill 2010), describes the relationship of heightened spiritual intimacy between more advanced *fuqara* and Shaykh Faye that revolves around the dispensation of knowledge.

While at the mosque's office where we kept most of our belongings, Abdur-Rasheed quietly showed me a special item in which Quranic inscriptions had been affixed to its front and back.[7] He revealed to me that he bought it from Shaykh Faye shortly before, and began to explain why he decided to pay so much money for the item. The protective nature of the item—which I will not name in order to maintain its confidentiality—was echoed by Shaykh Faye's explanation when he brought it up later amongst a small group of *fuqara* that afternoon during lunch at Umm Aisha's house. After our prayers, when everyone had settled and made themselves comfortable in a loose circle in the living room floor, Shaykh instructed Abdur-Rasheed on exactly how and when to utilize the item. While Shaykh Faye took care to teach those of us present the importance of spiritual protection, I also got the sense that there was more information that was shared privately with Abdur-Rasheed regarding his usage of the item. What Shaykh Faye did share with the group was that the item was essentially fashioned as a protective device and designed to bring wealth and health to the possessor and his family—it was an investment that one makes and can be inherited to loved ones.

### 4. Labor as Pious Formation and Embodied Knowing in Moncks Corner

In seeking to build upon Benjamin Soares' description of prayer economy, I would add, at least from what I have understood from this research, that the commodification of prayers and blessings shifts more than the relationship between teacher and student. There are specific ways in which the labor of students, and the ensuing performance of discipleship, acts as a commodity that operates in tandem to alter and enliven relationships in the *zawiyah*. Further, the internal logics surrounding such performances of labor are situated in discourses of personal spiritual transformation. At the same time, performances with regard to tea production, food preparation, or even salutations, mirror subtle social politics embedded in and around the community. This emphasis on disciplining the body as a strategy for cultivating the soul is one that is gained from a particular understanding of Islamic instruction in the *zawiyah*. Although students of Shaykh Arona Faye are expected to read the Qur'an and study other literatures for the purpose of improving their individual knowledge of their religion, this is not the major means of spiritual refinement. In this case, the Shaykh uses his own body as a means of instruction. Beyond witnessing impromptu lectures about the importance of hard (and consistent) work, students watch him spend his every waking moment devoted to service and care.

---

[7] Interview, Abdur-Rasheed W., 13 December 2014. The nature of the item will not be named here in order to keep the confidence and integrity of the item intact. There was not an explicit request to keep it private, but my commitment to scholarly integrity necessitates that the item and its process not be revealed so openly.

*4.1. Itinerant Labor as Spiritual Strategy*

During my time spent living at the Masjidul Muhajjirun wal Ansars in Moncks Corner, I had the opportunity to closely observe many "high" moments that occurred at the mosque (Friday congregational prayer, weekly evening classes, Saturday evening dhikr, Sunday afternoon classes, etc.). Of course, during programs and planned events where there were more people in the mosque, the halls were full of laughter and the warm embrace of brotherhood and sisterhood. In these moments, a large portion of the *fuqara* men and women attended the mosque to learn about their religion and share food; however, I also had an opportunity to observe more quiet, intimate moments when there was little activity. Making a living in Moncks Corner can be difficult, so many decide to pursue a career in the electrical trade and engage in contractual labor. In small groups, or individually, men within the community who were able and experienced applied for jobs in other locations. Some jobs involved installing electrical wiring as a journeyman or overseeing some project if the applicant was a licensed electrician. Most of these jobs required the men who accepted them to leave town for months at a time in order to earn money to send back to their households in Moncks Corner. It was this strategy of intermittent absences that made their general presence in Monks Corner, and proximity to the zawiyah, possible. Of course, many fuqara also did not have to leave to earn their living. Those who worked in town often started their own businesses: Imam Rasheed Nurridin ran a used car lot when he lived closer and has since opened a restaurant further away from town; Marjonah Ibrahim established herself as a masterful real estate agent in the area; Ishmael and Bilqis Nurridin opened a seafood restaurant and halal meat store; Muhminatou and Penda Sanno both ran an African hair-braiding salon; Halima and Sulaiman also opened another hair salon in town while engaged in other entrepreneurial activities. Some *fuqara* also work in nearby retail stores, run formal and informal childcare locations, buy and sell products imported from other countries, and some do odd jobs to get by. While many who are part of the community are originally from the area, many others have specifically relocated to Moncks Corner from other parts of the country, leaving potentially lucrative opportunities, in order to sit at the feet of Shaykh Arona Faye on a permanent basis. In order to sustain themselves, many *fuqara* have resorted to itinerancy as a strategy for spiritual growth. Thus, contractual electrical work within the community serves this purpose.

The preference for practical itinerancy and mobility over settlement seen amongst the *fuqara* is a mode of life that simultaneously betrays the assumption of an orientation toward urban environments and material advancement. The Muslims in Moncks Corner have, mostly, actively chosen to resist the allure of modern distractions and the infrastructural conveniences of city life. Read here as purposeful and pragmatic, the collective decision to have continual and sustained access to the *zawiyah* of Moncks Corner is driven by the desire to grow inwardly by sacrificing material gain. In a similar manner, Magnus Marsden (2009) resists simplifying Muslim mobilities in northern Pakistan as mere shifts from village life to urban environments or allowing for discourses of modernity to take a matter-of-fact placement in his analysis. His examination of these purposeful itinerancies offers an opportunity to view how young Muslim men, for example, vacate their urban lifestyles for the respite that the life of the guest (or student) has to offer. In particular, his approach is useful due to the manner by which his ethnography betrays an automatic reading of Muslim travel as cosmopolitanism in favor of being driven and shaped by locally embedded identities. I interpret, therefore, the choice to either relocate (or remain) in Moncks Corner as an intentional practice grounded in a concern for spiritual expansion and empowerment.

As well, there were men who willingly lived in the mosque (*Masjid Muhajjirun wal Ansar*) on a more than part-time basis and were charged with daily maintenance of the property. Due to their continual presence, these men were frequently on-call to travel with Shaykh Arona Faye to accompany him on trips to nearby religious programming where he (and his entourage) were received as guests of honor. In this way, the Muslim men who lived in the mosque enjoyed continuous access to Shaykh Faye as they were

present for every morning prayer and evening prayer, and had ready access to the multiple classes held per week at the mosque. Thus, itinerancy and, for some at least, material dependency were articulated as methods for sustained learning and spiritual cultivation among devotees in the Moncks Corner *zawiyah*. Moreover, many of the more devoted adherents within the *zawiyah* used their bodies as "currency" in this prayer economy. That is, the service they provided in the form of tea preparation or dictating Qur'anic verse for students and clients, whether temporary or sustained, was but another form of labor whereby participation contributed to the raising of funds and dispensation of knowledge throughout the network and beyond. Of course, it is not that these particular gentlemen were seen as somehow more righteous or pure than members who lived in town and could support themselves and their families through regular employment or other means. In fact, the opposite is probably more accurate. At times, some of the men who had habitually been in residence at the mosque for some significant time were lightly chided by Shaykh Arona Faye as he encouraged those who were single to get married and for those who seemed incapable of supporting themselves to work more regularly.

This detail should not, however, encourage us to view these men as simply listless and or indolent. I argue that at least some of the men with whom I lived labored intensely, although the manner in which they labored might not, at first glance, appear to be the case. The *fuqara* who lived in the mosque did so for numerous reasons—some were rebuilding their lives after incarceration, some were rebounding from failed marriages, some were preparing to relocate abroad to Senegal and chose to use the residence in the mosque as an anchor, and some were mere visitors. That is, they all seemed to be in the midst of some kind of laboring and in different ways. Their residence at the mosque often implied a heightened responsibility as far as cleaning and beautifying the mosque, securing the mosque premises, promptly calling the *adhan* (call to prayer) at its appropriate times, receiving out-of-town guests and visitors upon their arrival, and, in some cases, teaching Arabic classes when needed.

### 4.2. Umm Aisha Helps Birth the Moncks Corner Zawiyah

In spite of the fact that during more formalized interviews my interlocutors seemed to all articulate a desire to financially compensate Shaykh directly or somehow contribute funds to his mission, I have witnessed the manner in which most, if not all, students actively attend to Shaykh Faye's needs and directives in his presence and thus embody an ethos of hospitality and care. For example, the Moncks Corner community, as well as the many guests who have visited Masjid Muhajjirun wal Ansars, know that the women of the Moncks Corner zawiyah have collectively cared for the hundreds, if not thousands, of Muslims who have visited the community and have been doing so for the past twenty-five years. More importantly, however, is the sustained commitment on the part of the women to collaboratively feed the community every single week after the Friday congregational prayer. In addition, women like Umm Aisha have even devoted their houses to receiving students to be fed in intimate company with Shaykh Faye almost nightly. Almost throughout the entirety of my fieldwork, Umm Aisha's modest-sized house on Tall Spruce Street operated as an evening meeting place for the men who lived at the mosque, in addition to a select few, who were honored with cold juices, hot southern comfort foods, cakes and assorted deserts. Such reception was an example of unparalleled southern hospitality as an average of ten hungry bellies sat in an intimate circle around a tablecloth that was carefully placed on the living room floor. While Shaykh Faye taught those of us present by relating hadith and morally infused anecdotes to make his teachings more relatable, Umm Aisha labored intensely to ensure that we all were taken care of. Her efforts indirectly suggested the importance of dedication and consistent care for others. Her name "Umm Aisha" in this regard has been well-earned insofar as her mothering of the entirety of the community has been constantly recognized publicly by Shaykh Arona Faye. He acknowledges all that she has done for the community—and the Tariqa—due to her willingness to devote

her own property as well as her physical labor to support the presence of the Mustafawi movement in the United States.

The maintenance of a prophetic tradition (*sunna*) through caring for guests that is mediated by simultaneous West African values and Black American southern hospitality might also be interpreted as an act of resistance to Western social norms that emphasize individualism and anti-Muslim sentiment. Rouse and Hoskins (2004) describe how African-American Muslim women in southern California use their bodies as sites of resistance to American cultural and social norms that read the shedding of clothing to be a marker of freedom. More importantly, they find that these women negotiate multiple overlapping networks on an individual, as well as collective, level, through the preparation and dispensing of food. Keeping in line with the question of resistance that they bring up in their work, Rouse and Hoskins discuss the manner by which African-American Muslim women articulate complexities of race, gender, class, and faith. Not only do they participate in community space as wives and mothers, they serve a vital function as creators and sustainers of collective identity in racial, historical, and religious contexts. Rouse and Hoskins assert that:

> ... through food, female converts articulate their relationship to a number of ideological domains including race, class, gender, nation, and Islam. As a signifying practice associated with issues of race, authenticity, and group membership including citizenship; food preparation and exchange are vital communicative processes. Women who are generous with food and who understand the dietary requirements of the community, have extensive social networks and are credited with having a greater understanding of the faith. It is through food that women gain membership into various overlapping social networks, and it is through these social networks that women developed organized systems of exchange. Without these exchange networks, many of the women would not have sufficient income to pay rent ... The quality and preparation of food is about faith, ideology, community, and securing resources. Embodied in the production and eating of food is the performance of an agency owned not so much by individuals, but by a community intent on authorizing new social configurations. (Rouse and Hoskins 2004, p. 228)

Like others (McCloud 1995; Curtis 2006), Rouse and Hoskins inform us that the politics of consumption in African-American Muslim communities carries a historical genealogy that draws from the Nation of Islam's ideology of the body where the adoption of alternative nutrition underlined a posture of resistance that separated African-American Muslim identities primarily in urban centers of the northern United States from despised ideations of blackness in the postbellum South.

Joseph Hill frames such attentiveness to providing care through sustenance as "devotional cooking" in which female Sufi leaders (*muqadammas*) in the Senegalese Niassene Tijani tradition understand their culinary contributions as a medium for gaining blessings and prestige (Hill 2018). They understand their domestic labor as an opportunity to instruct students and embody the *tarbiyah* that reflects their own training; but, then again, care was the focus of Umm Aisha's professional career as well. She had relocated to South Carolina from up north in 1973 after receiving her license as a registered nurse. As the only black registered nurse who worked at the Medical University of South Carolina in Charleston, Umm Aisha developed an expertise in laboring on behalf of others. Her days in the Nation of Islam also emphasized this focus on care and she was encouraged to pursue midwifery as well once she had the opportunity to take classes at the Medical University of South Carolina (MUSC)—Charleston. Upon achieving that certificate, she was then the only black registered midwife in South Carolina. By the time she and Shaykh Faye were ready to initiate the Mustafawi mission in the United States, Umm Aisha had been well-positioned to care for a community of Muslims by using her own house as a *zawiyah*—indeed, what Shaykh Faye has repeatedly described as a "hospital".

During an interview in her house, Umm Aisha explained that at the time of her decision to marry Shaykh Faye she had been a woman who had her own finances due to her career as a registered nurse–midwife and did not require the monetary support of a man. Therefore, she fully supported the notion that Shaykh Faye's main job was to propagate Islam. After being married for about two years, Shaykh Faye finally was able to relocate to Moncks Corner in 1994. Initially, he taught American Muslims the specificities of Islamic creed and practice at a predominantly African-American mosque in Charleston that took its leadership and spiritual direction from the late Imam Warith-ud-Deen Muhammad (d. 2008). After the eruption of some "ideological differences", Shaykh Faye and Umm Aisha decided that it was time to open a mosque closer to home in Moncks Corner. After all, there were a few Muslims who lived in town such as Umm Zubaidah Gibbs and her two sons. So faithful was Umm Aisha to the idea that Islam should have a foothold in her town that she was willing to mortgage her own house to pay for a building on Carolina Avenue, which would become the first iteration of Masjidul Muhajjirun wal Ansar and the main *zawiyah*:

> " . . . I had prayed for a Muslim husband and I met [Shaykh Faye] . . . I know that his job was to propagate Islam and I felt that, I knew I had a good job and I had finances and I felt like, you know, when I met him I said 'I can do that, I can assist him in propagating Islam' . . . so that's where I started off as far as the idea of being married to a Muslim man . . . [8]

She has since retired from this practice, and the responsibility of hospitality has been entrusted to women such as Ndey Faye, eldest daughter of Shaykh Faye, who has relocated with Shaykh Mikhail, her African-American husband and protege of Shaykh Arona Faye, to Moncks Corner from Dakar. However, Shaykh Faye readily acknowledges that without the efforts of Umm Aisha, Masjidul Muhajjirun wal Ansars and the Muslim community of Moncks Corner would not be the bastion of light that members and visitors believe it to be. Their collaborative effort to provide care for a growing community is emblematic of the kind of mutual care seen amongst community members. Surely, Shaykh Faye has dispensed much to Umm Aisha; and she to him. Their cooperative labor has facilitated not only the growth of a Sufi community; their labor has also propelled them to grow as Muslims who live righteously and observe the traditions of Prophet Muhammadﷺ.

Using the language of *indebtedness* to describe the relationships among the *fuqara* of Moncks Corner, whether economic or labor-based, provides a manner for expanding the "prayer economy" as a theoretical field. Benjamin Soares' framing of material exchange in West African Muslim community offers an opportunity for thinking about the possible relations that extend beyond the community in which a prayer economy might function. In taking his insights further, I propose a consideration of how the transfer of gifts, blessings, and debts between Muslim actors—especially in a single Sufi tradition—are the result of prior transfers of blessings. That is, the blessings (*baraka*) and knowledge (*'ilm*) that circulate in the *zawiyah* of Moncks Corner do not originate in Moncks Corner. It has a genealogy and an ancestry, like Shaykh Faye, and it is most immediately attributed to Shaykh Muhammad Mustafa Gueye Haydara, founder and establisher of the Mustafawi tradition. As Soares articulates the manner in which prestige and charisma are built upon connections to personalities of import, prophetic genealogy, and spiritual expertise, my own framing seeks to unravel the manner in which charismatic religious specialists (e.g., *shuyukh*) become potentially indebted to others. While the conception of prayer economy, as revealed through Soares, places emphasis on the hierarchy and charisma of "free-floating sanctifiers", the economy of giving and labor framed here focuses on how the relation between Shaykh Faye and his students operates from within the context of a formalized Sufi tradition (Soares 2005, p. 153). Not only does he feel indebted to God for bestowing upon him the honor of conducting the mission to spread the religion of Islam, one might

---

8　Interview, Aisha Faye, 17 November 2014.

also say that Shaykh Arona Faye is indebted, in a way, to his students whose spiritual and material needs provide him the ability to dispense the knowledge transferred and entrusted to him via righteous ancestors. If knowledge is the lost property of the believer, then Shaykh Arona Faye has a duty to dispense that knowledge as best as he can and as widely as he can. One manner in which the Shaykh labors intensely is within realm of the secretive and, like those before him, he does so on behalf of his students. It is not so that the *fuqara* surround their Shaykh in order to serve his needs only—he also lives to serve their needs. Therefore, his engagement of secret sciences is embedded within a broader field of spiritual labor conducted for the benefit of his community.

Ben Soares' framing of prayer economy, and the efficacy of mystical power, hinges upon secrecy. The fact of the secret (*siru*, *sirr*) is what adds validity to the medicine or prayer of the charismatic shaykh. Surely, there is value contained in the lack of knowledge that others possess about the processual nature of the proper construction of amulets and talismans. Furthermore, he argues that a prayer economy is dependent upon an environment where "esoteric sciences" configure exchanges between Muslim actors (Soares 2005). Only the expert may know what and how certain Qur'anic verses or special prayers get placed into the signet ring, for example. For this reason (among others), the Shaykh maintains a charismatic allure that followers and admirers view as knowledge of the unseen. Esoteric sciences and secret knowledge, in the context of the Mustafawi, operate as a means of exchange for which American students are willing to labor as they are drawn into a West African paradigm of Islamic piety.[9]

It is in secrets that the power of the unseen (*al-ghaib*) operates. Shuyukh in West Africa have been a source where adherents and seekers of spiritual care turn to have their needs met as they are perceived to be the inheritors and possessors of the secrets that God gives. This confidence in mastery of the unseen is the basis upon which the Mustafawi are willing to compensate their own Shaykh Faye who has inherited his knowledge from Shaykh Mustafa. Moreover, the mastery of Shaykh Mustafa is accessible through the tutelage of Shaykh Faye and through the spiritual poems (*qasidas*) penned by Shaykh Mustafa himself. While Shaykh Mustafa is no longer physically present to display his own expertise, his writings reflect the way in which he was more than acquainted with the esoteric sciences. In a *qasida* written by Shaykh Mustafa Gueye Haydara, his composition illustrates a firm recognition of the power of the unseen. By appealing to Allah to grant him (and his students) the power to mold and impact the material world by way of that which is veiled from humankind, he seeks protection and success:

"1. By the truth of the secret of the opening chapter of the Book (Qur'an), give us all of our desires without any torment [ . . . ] 4. Give us the blessings of the noble descendants of the Prophetﷺ as well as their helpers (Al-Ansar), openly & in secret [ . . . ] 7. Grant us the sun of Irfan (intimate knowledge) & let the full moon of Your Nur (light) be our (inner) proof [ . . . ] 19. Grant us, in secret, the ease that comes with the subservice of Jinn & men [ . . . ] 31. Eliminate the harm of witchcraft, secretly & openly, from all places around us [ . . . ] 35. Make a way for us to Al-Mustafaﷺ, openly & in secret, O Custodian [ . . . ] 40. Remove the veils from our eyes & grant us the honor of being capable of providing comprehensive explanations. 41. Let our beloved ones & brethren testify for us, Ya Rabbana, in secret, with clear evidence. 42. Answer our prayers, O Owner! O Pure One! By the power of "Kun" (be) & "Yakun" (it is), O Bestower of benefit to the souls!

---

9   In a dissertation that historicizes the presence of "secret sciences" amongst Muslims in West Africa, Rupert Vaughn argues that the role of talismans and amulets are aspects of a broader tradition of engagement with Qur'anic scripture that highlights the perception of its spiritual power. "The talisman may be viewed as the slate upon which may be found expressions of the various forms of esoteric approach. Amulets provide a means by which good fortune, beneficence and protection may be garnered to the individual by means of siphoning, using and controlling [magical forces]" (Vaughan 1992, p. 45). Vaughn situates such emphasis on the use of amulets among the Mande who called amulets "*sebenev*" which meant "writing" in their language. This shows an association between scripture, or the act of writing scripture, and the physical product that results from the crafting of protective talismans. Further, he adds that "[t]he various forms of divination, magic and practices of an esoteric nature were all merely gateways through which an appreciation of al-asrar [secrets] could be gained" (Vaughan 1992, p. 46).

[ . . . ] 53. Grant us, O Possessor of power, to give things form, the gift of the ability to transform & change things from one to another [ . . . ] 55. Remove the veils of every sirr (secret) for us, Ya Rabbi, by the secret of the innermost secret [ . . . ] 58. Always be there for us extending Your helping hand, in secret & in the open, O Grantor of aid [ . . . ] 66. Cover our faults for us, Ya Rabbana, & grant us gifts from the unseen [ . . . ]"[10]

Shaykh Mustafa, in composing this *qasida*, clearly relies upon both the spiritual power of a living text and the subtextual force of a broader West African Islamic tradition in order to pray for money, the ability to effect immediate material change, and for Divine assistance and protection in all matters.[11] That the composition displays not only intimate knowledge of an otherworldly realm but also draws on Qur'anic text to build a forceful supplication is significant. The prayer is built from gnosis (*ma'rifah*), and not merely recall of Qur'anic scripture. It is from texts like this, in addition to the explanations of Shaykh Faye, that the *fuqara* feel confident in the esoteric mastery of their Tariqa's founder.

The path of inward transformation—of alchemy—occurs through training the physical body toward mindfulness, which is provided through the recitation of dhikr and through acts of attentive listening and pious labor. Therefore, an emphasis on embodiment is the key to understanding processes of knowing in (and beyond) West African Islamic context. "Human 'bodies of knowledge' are made, not born. Islamic learning is brought into the world through concrete practices of corporal discipline, corporeal knowledge transmission, and deeds of embodied agents. Knowledge of Islam does not abide in texts; it lives in people" (Ware 2014, p. 9). Such emphasis on the embodiment of knowledge—in other words, the performance of knowing—is similarly exemplified in the Mustafawi tradition. In order to be considered knowledgeable, one must embody piety and display righteous behavior. The attentiveness to Islamically ethical behavior and the mindfulness towards etiquette (*adab*) with regard to interpersonal relationships and worship are assumed to be a primary step in the pathway to knowing. Just as in Rudolph Ware's characterization of West African religiosity as a backdrop to Qur'anic memorization that places an emphasis on embodying the principles one memorizes through pious action, Muslims in Moncks Corner and in Senegal are expected to embody, or perform, the knowledge that Shaykh Arona Faye gives them. Following the mode of West African spiritual pedagogy, the Mustafawi tradition similarly necessitates an inseparability between knowledge and action. This is because the mode of spiritual training outlined here involves rectifying behavior and requires a corporeal modeling of piety learned from both present teachers and their ancestors.

## 5. Conclusions

The *fuqara* of Moncks Corner take seriously the notion that they are indebted to Shaykh Faye insofar as he has worked tirelessly to teach them Islam and show them an illuminated pathway to God. However, this sense of indebtedness is not imposed by the Shaykh himself. Feelings of obligation, it seems, are entirely self-imposed and reinforced by witnessing others in service to Shaykh Faye or providing funds for the mission of spreading Islam. Hence, the *fuqara* pay their debt to Shaykh Faye through maintaining self-discipline and following his guidance regarding private and public matters. In order to pay him back, they also use their hard-earned money and labor to support the mission of the Mustafawi. They understand that by helping Shaykh Faye, they help themselves. The path of the *faqir* is one that is preoccupied with letting go of material objects and piling up good deeds. It

---

10  Gueye Haydara, Shaykh Mustafa. Excerpts from "The Cloak of Protection & Soldiers of Divine Care". Translated and printed in the USA by Shaykh Arona Faye al-Faqir. Zawiyyah of Moncks Corner. August 2014.

11  The opening page of the printed qasida of Shaykh Mustafa Gueye displays the opening chapter of the Qur'an (*Al-Fatihah*) and is followed by the Mustafawi prayer that praises the Prophet Muhammad (*as-Salaat-ul Samawiyyah*). Below these two items is a brief explanation of the entire ballad: "This qasidah was written by Shaykh Muhammad Mustafa Gueye Haydar, who was the son of Shaykh Sahib Gueye, may Allah Ta´ala be pleased with both of them. With this qasidah he [beseeches] Allah (swt) to draw toward him all forms of goodness. He titled it "The Cloak of Protection & the Soldiers of Divine Care." He said that it would be exactly as [its] title suggests, in the open & in secret, for those who recite it morning & evening for the sake of Allah (swt) and with the intention of attracting all goodness and blessings repelling harm."

places emphasis on building spiritual expertise and inward refinement. They rely on their teacher to show them the way.

At the same time, Shaykh Faye is indebted to both his ancestors and his community. His grandfather and mother have bestowed upon him a legacy of learning, while his uncle and teacher, Shaykh Mustafa, has shown him an illuminated pathway upon which to lead others to God. Moreover, he has repeatedly exclaimed to the Moncks Corner community that it was surely indebted to Umm Aisha for her own selflessness insofar as she had given so much for the sake of seeing the community grow. During our interview, I remarked to Umm Aisha that the role she played in founding the *zawiyah* of Moncks Corner reminded me of the story of Hagar in the Arabian desert. Due to her faith, she was instrumental in not only financing the mosque in which the community would be sheltered, but she also was vital in feeding the community. Like Shaykh Faye, she was fully enveloped in the task of teaching others through provision. She was a vessel through which kindness and generosity were dispensed amongst the *fuqara* in Moncks Corner and beyond. As Shaykh Faye actively taught his students the basics of their religion and the route to a finer sense of spirituality, Umm Aisha modeled for others what the basics of religious etiquette and a finer spirituality looks like in motion.

**Funding:** This research was funded in part by the Center for African Studies and the Berkeley Center for the Study of Religion at the University of California at Berkeley.

**Institutional Review Board Statement:** The study was conducted according to the guidelines of the Declaration of Helsinki, and approved by the Institutional Review Board of the University of California-Berkeley 's Office for Protection of Human Subjects (CPHS# 2014-05-6380 / 30 May 2014).

**Informed Consent Statement:** Informed consent was obtained from all subjects involved in this study.

**Data Availability Statement:** Data sharing is not applicable to this article.

**Acknowledgments:** I would like to thank Shaykh Arona Faye, Umm Aisha, and the Muslim community of Moncks Corner, South Carolina for their patience and willingness to participate in this research. I also would like to thank Ousmane Kane, Mariane Ferme, and Rudolph Bilal Ware for their continuing encouragement and scholarly guidance along this journey.

**Conflicts of Interest:** The author declares no conflict of interest and external funding had no role in the design of the study; in the collection, analyses, or interpretation of data; in the writing of the manuscript, or in the decision to publish the results.

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
