# Peer review of "Fisibilillah: Labor as Learning on the Sufi Path"

_religions, doi:10.3390/rel12010003_

Round 1
Reviewer 1 Report
This article is a solid piece of ethnografic work, supported both theoretically and empirically. I am sure, it might be of interest for those dealing, in general and in particular, with Sufi traditions, the question of knowledge transmission, commodification of religious fenomena or mutual responsibility of the teacher and his followers in religious, especially Sufi, milieu.
I traced a few minor typos/misspellings easily correctable, e.g. line 47 (Every Sunday morningSuch); inconsistency of usage of italics ('Mustafawi Tariqa' and 'Tariqa' are written sometimes this and sometimes the other way, see e.g. lines 62, 84, 87, 101). Apart from that, I don't have any other remarks and recommend the text for publication.
Author Response
The typographical error that was pointed out has been addressed, and inconsistent usage of italics has been revised.
Reviewer 2 Report
So, this is a lovely, pious account of a particular Shaykh and his community. That's all well and good. But one has to anticipate the counter-arguments. I mean this with no disrespect to the Shaykh, or the author, as I don't know them. But when a Shaykh charges $5,000 for a journal, which he sells to 2 other followers, and the followers are people who travel and work as electricians, I wonder what percentage this is of their income. When the author is asked to go to an ATM to get cash for the Shaykh, one is suspicious. How is this man not a con-artist, living off the money of people who probably can't afford it?
Please know I am not MAKING these charges against the Shakyh. But there has to be SOME discussion of this, some refutation, some acknowledgment that outsiders might not have the same generous view that the author has. I'm not asking to be convinced that the Shaykh isn't a con-man, I'm just saying there needs to be some discussion of this in the paper.
Author Response
The reviewer has asked for some commentary that anticipates the notion of exploitation in the text. I have thus included some brief comments to address this. In the original submission, I did note that although money was freely given to the Shaykh, I actually witnessed the money and other valuable goods given to the less fortunate.
I have decided against any significant reframing in light of this, but offered additional commentary and several adjustments to hopefully make this more clear.